# Nanocomposite of Fullerenes and Natural Rubbers: MARTINI Force Field Molecular Dynamics Simulations

**DOI:** 10.3390/polym13224044

**Published:** 2021-11-22

**Authors:** Jiramate Kitjanon, Wasinee Khuntawee, Saree Phongphanphanee, Thana Sutthibutpong, Nattaporn Chattham, Mikko Karttunen, Jirasak Wong-ekkabut

**Affiliations:** 1Department of Physics, Faculty of Science, Kasetsart University, Bangkok 10900, Thailand; k.jiramatez@gmail.com (J.K.); w.khuntawee@gmail.com (W.K.); nattaporn.c@ku.ac.th (N.C.); 2Computational Biomodelling Laboratory for Agricultural Science and Technology (CBLAST), Faculty of Science, Kasetsart University, Bangkok 10900, Thailand; fscisrph@ku.ac.th (S.P.); thana.sut@kmutt.ac.th (T.S.); 3Thailand Center of Excellence in Physics (ThEP Center), Commission on Higher Education, Bangkok 10400, Thailand; 4Department of Material Science, Faculty of Science, Kasetsart University, Bangkok 10900, Thailand; 5Department of Physics, Faculty of Science, King Mongkut’s University of Technology Thonburi (KMUTT), Bangkok 10140, Thailand; 6Department of Chemistry, The University of Western Ontario, 1151 Richmond Street, London, ON N6A 3K7, Canada; mkarttu@uwo.ca; 7Department of Physics and Astronomy, The University of Western Ontario, 1151 Richmond Street, London, ON N6A 3K7, Canada; 8The Center for Advanced Materials and Biomaterials Research, The University of Western Ontario, London, ON N6A 3K7, Canada

**Keywords:** molecular dynamics simulations, natural rubber, *cis*-1,4-polyisoprene, MARTINI force field

## Abstract

The mechanical properties of natural rubber (NR) composites depend on many factors, including the filler loading, filler size, filler dispersion, and filler-rubber interfacial interactions. Thus, NR composites with nano-sized fillers have attracted a great deal of attention for improving properties such as stiffness, chemical resistance, and high wear resistance. Here, a coarse-grained (CG) model based on the MARTINI force field version 2.1 has been developed and deployed for simulations of *cis*-1,4-polyisoprene (*cis*-PI). The model shows qualitative and quantitative agreement with the experiments and atomistic simulations. Interestingly, only a 0.5% difference with respect to the experimental result of the glass transition temperature (T_g_) of the *cis*-PI in the melts was observed. In addition, the mechanical and thermodynamical properties of the *cis*-PI-fullerene(C_60_) composites were investigated. Coarse-grained molecular dynamics (MD) simulations of *cis*-PI-C_60_ composites with varying fullerene concentrations (0–32 parts per hundred of rubber; phr) were performed over 200 microseconds. The structural, mechanical, and thermal properties of the composites were determined. The density, bulk modulus, thermal expansion, heat capacity, and T_g_ of the NR composites were found to increase with increasing C_60_ concentration. The presence of C_60_ resulted in a slight increasing of the end-to-end distance and radius of the gyration of the *cis*-PI chains. The contribution of C_60_ and *cis*-PI interfacial interactions led to an enhancement of the bulk moduli of the composites. This model should be helpful in the investigations and design of effective fillers of NR-C_60_ composites for improving their properties.

## 1. Introduction

Natural rubber (NR), mainly consisting of *cis*-1,4polyisoprene (*cis*-PI) with high elasticity, is acknowledged to be virtually irreplaceable in applications such as tires and seals. However, the neat NR usually suffers from poor mechanical strength [1,2]. To alleviate this problem, reinforcing fillers are widely used to improve the mechanical properties of NR materials. Impressive enhancements in the properties, including mechanical [3,4], optical [3,5,6], self-healing [7,8], electrical [4,6,9], thermal [4,6], rheological [4,9], and glass transition [10,11], have been reported. The mechanical properties of NR composites depend on many factors, such as the types, dimensions, concentrations, dispersion states, and alignments of the fillers [4,12,13].

Among the most important methods of tuning the properties of NR are the vulcanization and reinforcement by composites, in particular carbon black (CB) [14,15] Here, the focus is on the latter. The reinforcement of NR composites by carbon black (CB) has been intensively studied and broadly applied in truck and car tires [16,17]. CB may improve tensile properties [18], fatigue resistance [19], stiffness, and wear resistance [20]. The CB concentration has a significant influence on the rubber composite properties; e.g., increasing the amount of CB has been shown to lead to enhanced tear strength, tensile strength, and elastic modulus [21,22]. The study of Wang et al. [23] revealed that the piezoresistivity of CB-NR composites monotonically decreases with decreasing CB concentration. However, CB-reinforced nanocomposites at high loading have shown the decay of the storage modulus at high deformation, indicating high hysteresis [24,25]. This is called the Payne effect, and it has limited the application of CB-reinforced nanocomposites to tires. Moreover, CB is produced from petroleum feedstock, which has a negative impact on the environment [26].

NR composited with carbon-based nanoparticles, e.g., carbon nanotubes (CNT) or graphene, offers the possibility to produce lightweight composites with enhanced electrical and magnetic properties and thermal conductivity [4,27,28]. For example, it has been shown that a small amount of multiwalled carbon nanotubes (MWCNTs) at 0.5 parts per hundred (phr) in NR-MWCNTs composite could enhance the tensile strength, tensile modulus, and tear strength by 61%, 75%, and 59%, respectively [29]. Fullerene(C_60_) has been extensively used as a filler in polymer composites, especially for poly(3-hexylthiophene) in polymer-based solar cells for low-cost power production [30,31,32], and, in combination with epoxy, it has been shown to result in enhancements in the mechanical properties, electrical conductivity, flame resistance, and anti-corrosion [33]. In addition, the presence of fullerenes in NR composites has been shown to increase the modulus at 100% (*M*_100_), the modulus at 300% (*M*_300_) of elongation, and the shore hardness [34]. It has also been reported that the addition of 0.75 phr of fullerene in NR increases the *M*_100_, *M*_200_, *M*_300,_ and hardness by 178%, 137%, 113%, and 20.5%, respectively, compared to composites at 0.0065 phr of fullerene [35]. After aging (heating at 70° C for 168 h), Jurkowska et al. found an increase in the elastic modulus for [C_60_] < 0.5 phr, and, interestingly, the elastic modulus decreased at [C_60_] = 0.75 phr. That could substantially reduce the degradation effects of rubber composites [35].

Atomistic molecular dynamics (MD) simulations can provide nanoscale information, such as the dispersion and aggregation of fillers [36], segmental dynamics [37], polymer chain alignment [38], molecular interactions [36,37,39], and filler size effects [40]. Guseva et al. [37] investigated the macroscopic properties of non-crosslinked *cis*-PI films confined with silica substrate using atomistic MD simulations. Liu et al. [41] reported that the glass transition temperature (T_g_) of the *cis*-PI film increased when the film thickness was decreased. They also reported that, compared to a pure polymer system, the T_g_ increased when the filler was added. Raffaini et al. [42] studied rubber–fullerene conformations and observed fullerene aggregation when rubber chains were modified with COOH-termination. Although atomistic MD has been widely used in investigations of reinforced polymer nanocomposites, such simulations are computationally expensive. To overcome this limitation, coarse-grained (CG) models have been developed to produce simplified representations of polymer molecules to access much larger time and length scales.

One of the earlier works on the simulations of polymer melts was the seminal paper by Kremer and Grest focusing on melts well above the glass transition temperature [43]. An important step toward coarse-graining was taken in the 1995 article of Forrest and Suter, who showed how softer pairwise interactions emerge from an atomistic model upon averaging over the pair distribution [44]. The following years were very active for the development of CG models for polymers and polymer melts, including the works of Akkermans and Briels, who constructed a CG model with a focus on the preservation of thermodynamic properties [45], followed by a number of studies on the systematic coarse-graining of polymers and melts [46,47], modification of the dissipative particle dynamics (DPD) method for melts [48], and fluctuating soft-sphere models [49] with the aim of improving the accuracy of CG polymer melt modeling. Regarding the interactions of fullerenes and polymers, Huang et al. [50] developed a polymer-fullerene CG model for polymer-based solar cells, and Volgin et al. [51] investigated the diffusion of fullerenes using a CG model. Although there is no lack of CG models, the MARTINI force field [52,53] has become by far the most dominant one, and it has been used for simulations of amino acids, water, phospholipid membranes, fullerenes, some polymers, and RNA [54,55,56,57,58,59,60,61].

Recently, we introduced a MARTINI CG model for *cis*-PI to study natural rubber [62]. The interactions of the *cis*-PI and fullerene, however, were not included in the model. That is done in the current work by combining the CG models of *cis*-PI and fullerene to study *cis*-PI-C_60_ composites. The parameters are based on the MARTINI force field version 2.1 [52]. The thermodynamic and mechanical properties of the *cis*-PI-C_60_ composites were calculated and compered with the experiments and atomistic MD simulations. The influence of the C_60_ concentrations on the structural, mechanical, and thermal properties of the composites was analyzed. The results show that the interfacial interactions between the C_60_ and *cis*-PI play an essential role in the composites’ mechanical properties.

## 2. Methodology

### 2.1. Molecular Dynamics Simulations

In this study, *cis*-PI in melts and *cis*-PI composites at different C_60_ concentrations were investigated by coarse-grained (CG) molecular dynamics (CGMD) simulations. The CG model for *ci*s-PI was developed based on the MARTINI force field version 2.1 (University of Groninge, Groningen, The Netherlands) and is described in detail in Appendix A. A coarse-grained or superatom bead was used to represent an isoprene monomer. CG mapping of stretching and bending parameters for the *cis*-PI beads from united-atom MD trajectories were published in our previous work [62]. All systems consisted of 500 chains of 32-mers *cis*-PI, which is equivalent to 16,000 CG beads in total (an isoprene monomer was mapped into one bead in the CG model).

C_60_ molecules were randomly placed into the *cis*-PI melts with the number of 53, 107, 213, and 427 molecules corresponding to concentrations of 4, 8, 16, and 32 parts per hundred of rubber (phr), respectively. The details of all simulations are shown in Appendix A. The CG fullerene model for the C_60_ molecule called F16 was taken from Ref. [63]. After steepest descent energy minimization, CGMD simulations were performed using the GROMACS 5.1.1 (University of Groningen, Royal Institute of Technology, Groningen, The Netherlands) package [64]. The number of particles, temperature, and pressure were kept constant (NPT ensemble). *cis*-PI and C_60_ molecules were thermostatted separately at 300 K using the Parrinello-Donadio-Bussi velocity rescale algorithm with a time constant of 30 ps [65,66]. Pressure was held constant at 1 bar by the Parrinello-Rahman barostat [67] with a time constant of 1 ps and compressibility of 4.5 × 10^−5^ bar^−1^. The reaction-field method was used for the long-range electrostatic interactions [68,69], Lennard-Jones interactions were cut off at 1.1 nm, and periodic boundary conditions were applied in all directions. The simulation protocol had been tested and used in several prior studies [38,70,71,72,73]. All production simulations were run with time step of 30 ps for 21 µs. To confirm that equilibration had been reached, time evolution for the end-to-end distance and the radius of gyration were calculated (see Appendix A). The last 5 μs were used for data analysis and errors were estimated using standard deviation (SD). Molecular visualizations were done using the Visual Molecular Dynamics (VMD) software (University of Illinois Urbana-Champaign, Urbana and Champaign, IL, USA) [74].

### 2.2. Calculation of Thermal and Mechanical Properties

Glass Transition Temperature (T_g_)

The final states of the simulations after 21 µs were used as the initial structures to calculate the glass transition temperatures (T_g_). The systems were cooled from 300 K to 100 K with 10 K interval. The cooling rate was 0.1 K.ns^−1^. To equilibrate the systems, additional 500 ns were simulated at each temperature in the NPT ensemble. T_g_ was calculated using the intersection of the two lines describing the behavior at low and high temperatures on density-temperature (*ρ*-T) curves [75,76,77]. Density versus temperature curves for *cis*-PI in melts and composites are shown in Appendix A.

The thermal volume expansion coefficient (*γ*), specific heat capacity (*c*_p_), and bulk modulus (*κ*) were calculated from the fluctuations of the simulation box volumes (*V*) and enthalpy (*H*) using:(1)$$\gamma =\frac{\langle VH\rangle -\langle V\rangle \langle H\rangle }{N_{A}k_{B}T^{2}\langle V\rangle }$$
(2)$$c_{p}=\frac{\langle {\left(H-\langle H\rangle \right)}^{2}\rangle }{mN_{A}k_{B}T^{2}}$$
(3)$$\kappa =\frac{k_{B}T\langle V\rangle }{\langle {\left(V-\langle V\rangle \right)}^{2}\rangle }$$
where *k*_B_ is the Boltzmann constant, *N*_A_ Avogadro’s number, and *m* is the total mass of the system in atomic mass units. The angular brackets refer to averaging over simulation time.

### 2.3. Calculation of Solvation Free Energy

Thermodynamic integration (TI) [78] was applied to calculate solvation free energies of C_60_ in water, pure *cis*-PI, and *cis*-PI-C_60_ composites. The last frame from each of the 21 µs simulations was used as the initial structure. A C_60_ was added into the system and used as a coupling molecule. After steepest descent energy minimization, the MD simulations with the coupling parameter (λ) between 0 and 1 with interval of 0.05 were performed under the NPT ensemble. All systems were run for 300 ns, and the last 100 ns was used for analysis. The Bennett acceptance ratio (BAR) was applied to estimate the solvation free energy [79]. The solvation free energies are shown in Appendix A. The effect of C_60_ concentration is discussed in Section 3.4 (Figure 5).

## 3. Results and Discussion

### 3.1. Coarse-Grained Model Based on MARTINI Force Field of cis-1,4-Polyisoprene

The macroscopic and structural properties of the *cis*-PI in the melts were analyzed and compared to the experiments and atomistic simulations in order to validate the CG model (Table 1). The system density was found to be in good agreement, with the values being about 19% and 27% higher than in the experiments [80] and previous united-atom MD (UAMD) simulations [38], respectively. The thermal expansion coefficient was found to be 52% and 19% higher than in the experiments [81,82] and UAMD [38], respectively. In addition, the squared end-to-end distance (*R*_0_), the squared radius of gyration (*R*_*g*_), and polymer expansion factor (< $R_{0}^{2}$ >/< $$R_{g}^{2}$$ >) of the *cis*-PI in the melts are within 5% from the UAMD data [38]. *R*_0_ is defined as the average distance between the first and the last CG bead of the *cis*-PI chain. *R*_*g*_ was computed using.
(4)$$R_{g}^{2}=\frac{1}{N}\sum_{i=1}^{N}{\left(r_{i}-r_{cm}\right)}^{2}$$
where *r_i_* is the position of particle i and *r_cm_* is the position of the center of mass of the cis-PI chain. However, the bulk modulus was 3.8-fold lower than in the experiments [83] and 2.6-fold below the previous UAMD simulations [38]. One of the major contributors to this decrease of bulk modulus is the reduction in the degrees of freedom upon coarse-graining monomers into superatom beads, which diminishes the friction between branching atoms and changes entropy [84,85]. Higher mobility of the CG chains causes large fluctuations of the simulation box volume compared to the UAMD simulations [38].

### 3.2. Effect of C_60_ Concentration on Macroscopic and Structural Properties of cis-PI Composites

Adding C_60_ into the *cis*-PI composites leads to a linear increase in density (Figure 1a) due to the mass of the C_60_ filler. Figure 1b shows the enhancement of the bulk modulus as a function of the C_60_ concentration. A slow increase was found for [C_60_] < 16 phr (2.17 × 10^−4^ GPa per 1 phr), followed by a rapid increase at [C_60_] = 32 phr (4.81 × 10^−4^ GPa per 1 phr). The increases in the density and bulk moduli as a function of the C_60_ concentration is similar to previous UA simulations [86]. In addition, the thermal expansion coefficient (*γ*) and specific heat capacity (*c*_p_) at 300 K were computed using Equations (1) and (2). In the composites, both quantities increased when the amount of C_60_ increased (Figure 1c,d), similar to what has been observed in previous MD [87] and experimental studies [88] of polymer carbon-based nanocomposites.

The structural changes of the *cis*-PI chains at different C_60_ concentrations were also investigated by monitoring the *R*_0_ and *R*_*g*_. The averages of the *R_0_* and *R_g_* for the last 5 μs are provided in Table 2. For the *cis*-PI chains, they increased slightly upon increasing the amount of C_60_. The polymer chain expansion factor (< *R*_0_^2^ >/< *R_g_*^2^ >) of the *cis*-PI in the melts was 6.11, which is in good agreement with previous UAMD simulations (6.20 [38] and 5.44 [77]) and refers to a random coil structure of a *cis*-PI chain [89].

### 3.3. Effect of C_60_ Concentration on Microscopic Properties of cis-PI Composites

#### 3.3.1. Interaction of C_60_-C_60_ and C_60_-*cis*-PI

To investigate the interaction free energy of C_60_-C_60_ and C_60_-*cis*-PI in the composites at different C_60_ concentrations, the potential of mean force (PMF) was calculated from the radial distribution function (RDF; *g*(*r*)) [90,91] as:
(5)$$\mathrm{PMF}=-k_{B}T\ln\left(g\left(r\right)\right)$$
where $k_{B}$ is the Boltzmann constant and T is the absolute temperature.

The dimerization free energies of C_60_ in *cis*-PI composites at different C_60_ concentrations are shown in Figure 2a. The first, second, and third minima of the C_60_-C_60_ PMF corresponding to the distances between the neighboring C_60_ molecules are at 1.0 nm, 1.5 nm, and 1.9 nm, respectively. The second minimum is at the lowest free energy, suggesting energetic preference of C_60_ dimerization with the *cis*-PI chain insertion. That the polymer chains fill the space between the C_60_ molecules has also been observed in previous studies [38,59,60,92]. The PMF for C_60_ and *cis*-PI interactions is shown in Figure 2b. This result demonstrates the C_60_-*cis*-PI composites’ preference to a dispersed structure corresponding to the snapshots in (Figure 3), which show the dispersed structure for all the C_60_ concentrations.

In Figure 2a,b, the first and second local minima are at 0.8 nm and 1.2 nm, respectively. Regarding the first local minima in Figure 2a,b, the free energy for C_60_-*cis*-PI is lower than for C_60_-C_60_, indicating a preference for the C_60_-*cis*-PI interaction. Moreover, in the case of C_60_-C_60_ interaction, a concentration dependence was observed in which the free energy decreased slowly upon increasing the C_60_ concentration from 4 to 8 phr. For the C_60_-*cis*-PI, only a minor concentration dependence is present.

#### 3.3.2. Diffusion of C_60_ and *cis*-PI in the Composites

The diffusion coefficients of the C_60_ and *cis*-PI in the melts and in the composites were analyzed at different C_60_ concentrations using the mean squared displacement (MSD)
(6)$$MSD=\langle r^{2}\left(t\right)\rangle \;\sim \;6Dt\;$$
where *D* is the diffusion coefficient and *t* is time.

In agreement with previous studies [38,87], the presence of C_60_ caused a dramatic decrease in both the C_60_ and *cis-*PI diffusion coefficients (Figure 4). The slow movement at high C_60_ concentration led to confinement, with the C_60_ molecules being more mobile than the *cis*-PI ones. The difference between the diffusion coefficients of C_60_ and *cis*-PI decreased upon increasing the C_60_ concentration; the increased interactions between C_60_ and *cis*-PI cause a reduction in the diffusive motions of C_60_ and *cis*-PI, which is the reason why the bulk modulus of the composites at high C_60_ concentrations becomes enhanced. Note that the diffusion coefficients, extracted from the CG simulations, were one order of magnitude faster than those obtained from the UA simulations [38]. This is due to the reduction of degrees of freedom in the CG model [93].

### 3.4. Effect of C_60_ Concentration on C_60_ Solvation Free Energy

The solvation free energies of C_60_ in water, *cis*-PI melt, and *cis*-PI-C_60_ matrix composites were estimated by thermodynamic integration (TI) [78]. The solvation free energy of C_60_ in water was −99.53 ± 0.03 kJ/mol, which is in good agreement with our previous CGMD calculation using umbrella sampling (−92.6 kJ/mol) [94]. Figure 5 shows the concentration dependence of the solvation free energy. The lowest value (~−115.53 kJ/mol) was found when an added C_60_ was solvated in a *cis*-PI melt. It indicates that the *cis*-PI melt is more favorable than water. Increasing the amount of C_60_ concentration led to lesser solvation in the composites. Our results suggest that the C_60_ prefers to interact with the *cis*-PI chains rather than the other C_60_ molecules in the *cis*-PI-C_60_ composites (Figure 2a,b).

### 3.5. Effect of C_60_ Concentration on Glass Transition Temperature (T_g_)

As discussed above, the CGMD simulations of *cis*-PI-C_60_ composites using our model are able to reproduce the results from previous experiments and atomistic MD simulations [50,95]. The glass transition temperature (T_g_) is one of the most important parameters determining the physical state and final mechanical properties of polymer composites [96]. To gain more understanding on the influence of the filler concentration on composites’ properties, T_g_ was estimated using the change of slope in density (*ρ*) versus temperature (T) curve [75,76,77]. The *ρ*-T curves of the *cis*-PI in the melts and composites are shown in Appendix A. The calculated T_g_ was 201 K, in good agreement with the all-atom (209 K) [77] and united-atom models (223 K) of Sharma et al., [77], and experimental data (200 K) of Brandrup et al. [82] Finally, the concentration dependence of the T_g_ in the composites was determined. The data in Figure 6 show a linear increase with a slope of 0.63 K/phr. This same trend has been reported in previous experiments [34] and simulations [41,87].

## 4. Conclusions

We have developed a MARTINI force field version 2.1-based [52] parameterization and performed systematic studies of *cis*-PI in melts and *cis*-PI-C_60_ composites using it. We validated the thermodynamic, macroscopic, and microscopic properties of the *cis*-PI in the melts, and the results show good agreement with prior experimental [80,81,82,83] and computational studies [38,77,94]. Perhaps most surprisingly, only a 0.5% difference with respect to the experimentally found glass transition temperature (T_g_) [82] in the *cis*-PI in the melts was found.

The properties of the NR-fullerene composites at different C_60_ concentrations (0, 4, 8, 16, and 32 phr) were studied by using CGMD simulations over 200 microseconds. The density, bulk modulus, thermal expansion, heat capacity, and glass transition temperature increased upon increasing the C_60_ concentration. A slight increase in the R_0_ and R_g_ of the *cis*-PI chains was found for the NR-C_60_ composites. The interaction energies between the C_60_ and *cis*-PI decreased when the amount of C_60_ increased. The diffusion of the *cis*-PI and C_60_ was slowed down by increasing the C_60_ concentration. The decrease of the interaction free energies and the diffusion coefficients resulted in an enhancement of the bulk modulus of the composites. Moreover, the solvation free energies of the C_60_ in the *cis*-PI matrix composites increased upon increasing the C_60_ concentration. These results suggest that C_60_-*cis*-PI interactions are more preferable than C_60_-C_60_ self-interactions.

The CG model introduced here allows simulations of large systems over long periods of time to study properties such as the impact of the filler concentration on rubber composites at the molecular level. This validated *cis*-PI-C_60_ CG model based on the MARTINI force field will also enable simulations of the advanced rubber materials and their interactions with biological molecules.

## Figures and Tables

**Figure 1 polymers-13-04044-f001:**
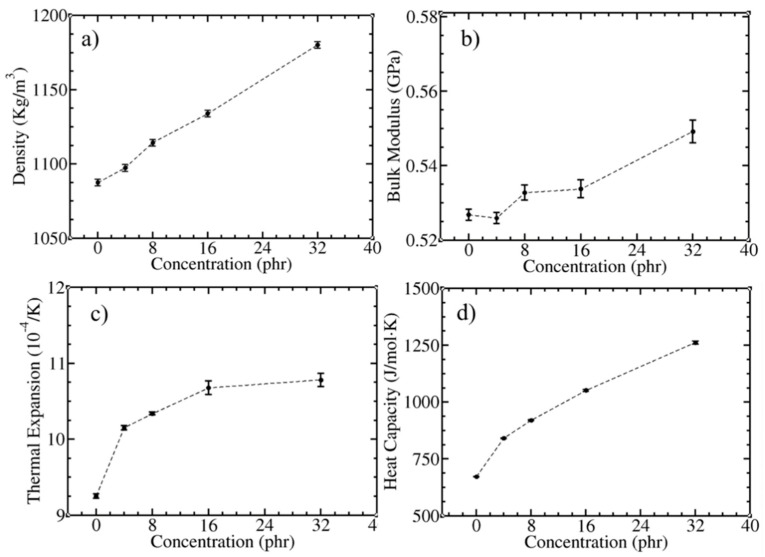
(**a**) Density, (**b**) bulk modulus (Equation (3)), (**c**) thermal expansion (Equation (1)), and (**d**) heat capacity (Equation (2)) of *cis*-PI and C_60_ composites as a function of C_60_ concentration.

**Figure 2 polymers-13-04044-f002:**
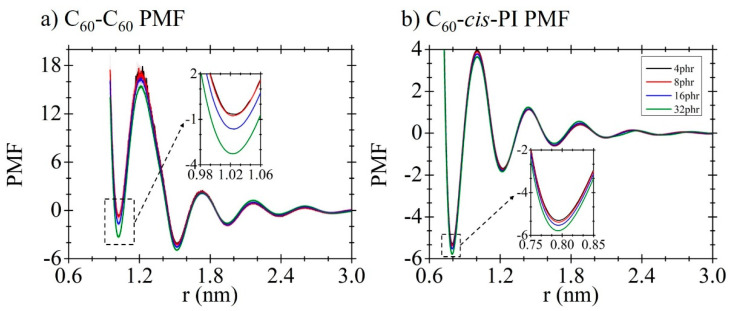
Potential of mean force (PMF) of (**a**) C_60_-C_60_ and (**b**) C_60_-*cis*-PI in the composites at different C_60_ concentrations.

**Figure 3 polymers-13-04044-f003:**
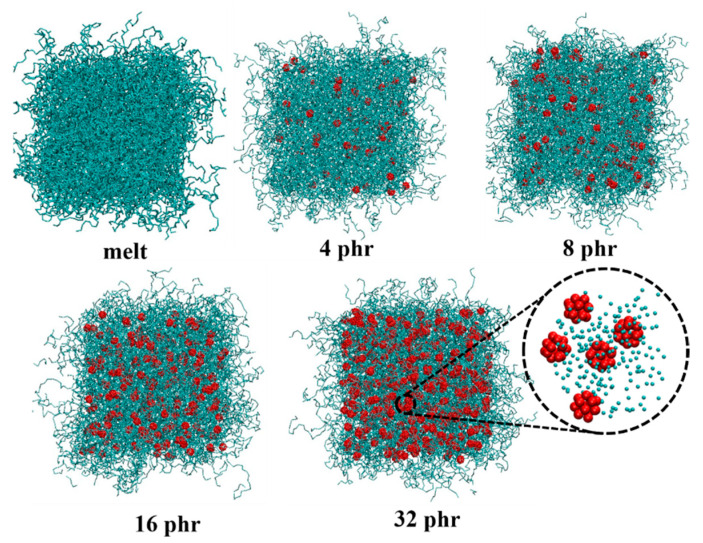
Snapshots of *cis*-PI-C_60_ composites at different C_60_ concentrations from 0 (melt) to 32 phr. Cyan and red represent *cis*-PI and C_60_, respectively.

**Figure 4 polymers-13-04044-f004:**
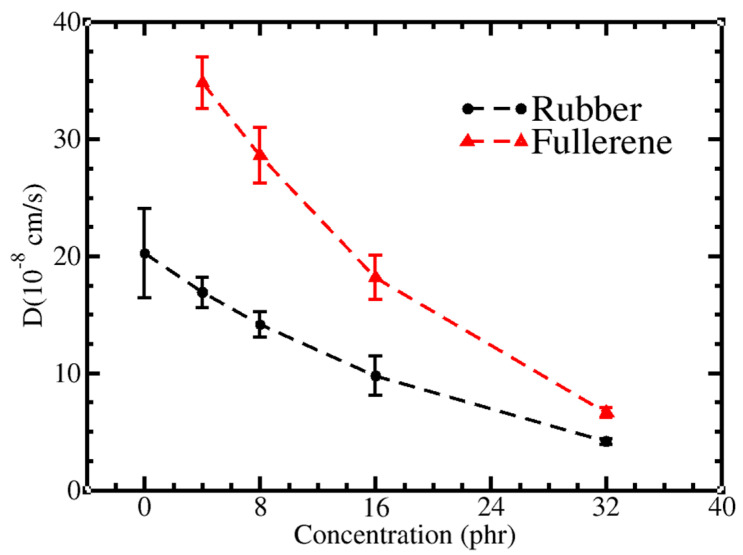
Black line: the diffusion coefficients of rubber in melt (0 phr) and composites (phr ≠ 0). Red line: the diffusion coefficients of C_60_ in composites as a function of C_60_ concentration.

**Figure 5 polymers-13-04044-f005:**
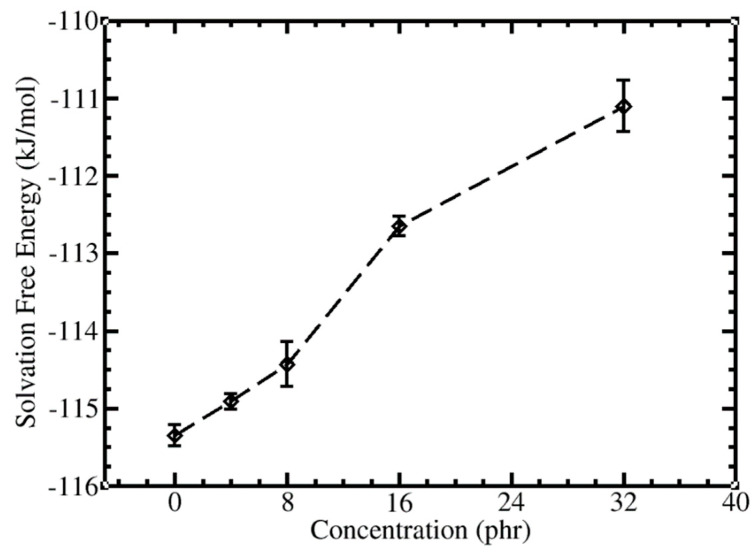
Solvation free energies of C_60_ in *cis*-PI melt (0 phr) and *cis*-PI-C_60_ composites at different C_60_ concentrations. The dashed line is a guide to the eye.

**Figure 6 polymers-13-04044-f006:**
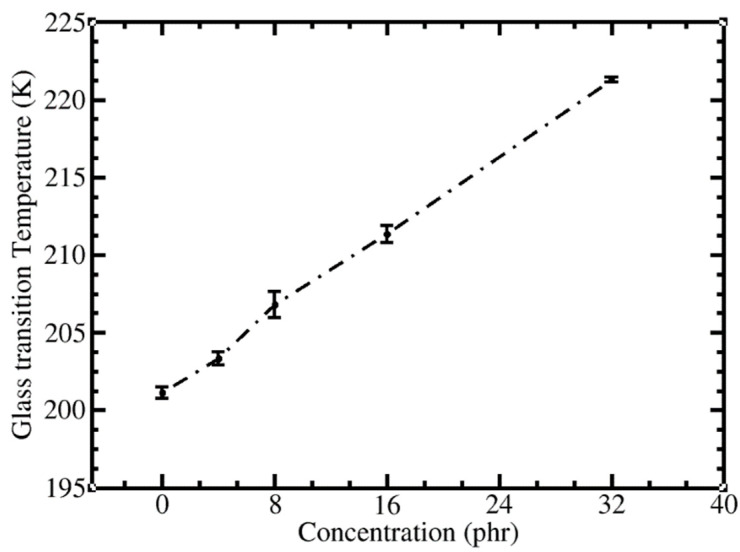
Glass transition temperature (T_g_) of *cis*-PI-C_60_ composites at different C_60_ concentrations.

**Table 1 polymers-13-04044-t001:** Macroscopic and structural properties of *cis*-PI in melts. *R*_0_, *R_g_*, and $$<R_{0}^{2}>/<R_{g}^{2}>$$ are the end-to-end distance, the radius of gyration, and polymer expansion factor, respectively.

Properties	Exp.	United Atom [38]	CG
Density; kg/m^3^	910 [80]	853.9 ± 1.6	1087 ± 2.30
Bulk modulus; GPa	2.02 [83]	1.37 ± 0.02	0.53 ± 0.00
Thermal expansion; (10^−4^/K)	6.1 [81,82]	7.80 ± 0.15	9.25 ± 0.02
< $$R_{0}^{2}$$ >	-	12.85 ± 0.41	13.28 ± 0.06
< $$R_{g}^{2}$$ >	-	2.07 ± 0.04	2.17 ± 0.01
$$<R_{0}^{2}>/<R_{g}^{2}>$$	-	6.20	6.11

**Table 2 polymers-13-04044-t002:** Average end-to-end distance (*R*_0_), radius of gyration (*R*_*g*_) of *cis*-PI chains, and their polymer chain expansion factor ($$<R_{0}^{2}>/<R_{g}^{2}>$$) as a function of C_60_ concentration.

C_60_ Concentration (phr)	< *R*_0_ > (nm)	< *R*_g_ > (nm)	$$<R_{0}^{2}>/<R_{g}^{2}>$$
0	3.64 ± 0.06	1.47 ± 0.01	6.11
4	3.65 ± 0.06	1.48 ± 0.01	6.11
8	3.66 ± 0.06	1.48 ± 0.01	6.12
16	3.67 ± 0.06	1.48 ± 0.01	6.12
32	3.71 ± 0.06	1.50 ± 0.01	6.15

## Data Availability

The data presented in this study are available on request from the corresponding author.

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
