# Peer review of "Nanocomposite of Fullerenes and Natural Rubbers: MARTINI Force Field Molecular Dynamics Simulations"

_polymers, 2021, doi:10.3390/polym13224044_

Round 1

Reviewer 1 Report

This paper systematically studied fullerenes-natural rubber composite materials including forcefield development, thermodynamical / dynamical characterization, and the fullerene content effect on natural rubber properties. The comparison with experimental data and united atom model validated the coarse grained forcefield and the property matching results are impressive considering degree of freedom reduction of MARTINI forcefield. The development of such a coarse-grained model for the purpose of studying large scale and long-time simulation could be quite important with many potential applications in the future. The manuscript is written in an easy-to-follow style and introduction part provide enough information covering previous and related study in this application. I would like to recommend it to be published on Polymers after the following major revision:

  • The supplementary information is missing, and I cannot find it anywhere on the reviewing website or inside the manuscript. The authors should include it in the reviewing process either attaching it to the main text or upload separately. Some information is missing such as Table S1/S2, Fig S1 etc. If I was right, the forcefield parameters and simulation details (which is quite important for reviewer to validate and reproduce the simulation results) are also missing during the review process.
  • I am curious about how the authors obtained the forcefields based on MARTINI force field. The property fitting looks really good, and did the authors use any special algorithms during the parameterizing process. I even don’t know what type of coarse-grained particle you are using in this model. So please include the detail information in the SI and make it clear for reader to follow and reproduce your computation results.
  • Snapshot of the simulation system and the comparison with atomistic and experimental photos should be included in either the main text or the SI. The aggregation state of Fullerene inside the rubber matrix could be quite important for the mechanical property of the composite. Is there any phase separation of the fullerene observed during the simulation? From Fig.2, it looks like the C60-C60 PMFs minimal position did not change too much when the concentration varied from 4 phr to 32 phr. If there is no C60-C60 aggregation, or phase separation, C60-C60 distance peak should be changing as the concentration increasing. From the results shown in Fig. 2, what I can read is somehow the C60 and cis-PI chain form a sandwich structure and C60s are glued by the cis-PI single chain. Therefore, C60 still prefers an aggregation state in a bigger picture. Could the authors add some comments on that?
  • All equations should be labeled with number, I can see after some equations, “eq. X (X)” is used, which is redundant. Eq. 5 also miss the equation number.
  • At the end of the second paragraph of the Conclusions section, the authors mentioned C60-C60 interactions are more preferable than C60-C60 self-interactions, which should be a typo according to the results shown by the authors.

Author Response

Dear reviewer,

The manuscript and supporting information were revised according to your suggestions. Changes made in the revised manuscript were shown in red. We would like to acknowledge useful suggestions, which certainly helped to increase the quality of in the present form manuscript.

Reviewer 2 Report

The authors present a MARTINI Force Field Molecular Dynamics simulation about nanocomposite of fullerenes and natural rubbers. This is an interesting paper; however, we have some concerns concerning presentation of the results.  

1.In the Abstract it is written that “The model shows qualitative and quantitative agreement with experiments and atomistic simulations.”, but in the text it turns out that this agreement means „ Thermal expansion coefficient was found to be 52% and 19% higher than in experiments [79,80] and UAMD [36]”. I think that 52% difference as good result should be explained.

  1. In the manuscript there are references to Tables S1, S2, S4 and Fig S1, but I did not find any Supplement or reference to its availability.

The article cannot be judged in its present form. I suggest publication of the manuscript only if the authors clarify the above-mentioned remarks.     

Author Response

(The authors gave the same response as above.)

Reviewer 3 Report

  the authors extend the previous study to investigate the effect of C60 content on the modulus, thermodynamic and other properties of natural rubber(NR). This study is quite relevant and consistent with many recent researches. However, the presented work has many shortcomings in terms of its technical correctness and result analysis. Considerable improvements are required for the study to be suitable for publication in journal of Polymers. please go though the following changes.

  1. It is well known that vulcanization is one of the most important reasons for the expected properties of NR. However, it is clear that the effect of vulcanization has been neglected in this study, which does not fit well with the context of most references in the INTRODUCTION. If this study focused on the NR melt interaction with fillers, should some studies with higher relevance to NR melt be cited to make the introduction more relevant to the topic of study?

2.The simulation methodology needs to be revised with more detail, which enable other scholars to replicate, extend and better understand the study. A few suggestions are as follows.

  1. a) Energy minimization and equilibrium are mentioned on page 3, which are essential steps in MD simulation. Therefore the criteria for reaching equilibrium need to be clarified. (based on energy amplitude or density amplitude? How much is the amplitude? Or other basis?)

  1. b) This study is an extension of the previous study[60]. The method and parameters of the coarse-grained NR molecular chain have been well expressed in the previous work. However, the coarse-grained C60 is not mentioned in the previous work and in this manuscript. It is recommended to add its bead details.

  1. c) The text form is adopted for the expression of all related model establishment in this manuscript, which is not concretely. In addition, dispersion of filler can clearly observed though configuration of the models, which is an advantage that text expression does not have. Therefore, it is recommended to use certain diagrams to express the construction of the model.

3.The authors have calculated and analyzed many properties of the models, but some of them have made me doubt, mainly summarized as the following points.

  1. a) In section 3.1, the results of this study are compared with other researches. What is worrying is that only the density parameter is relatively close among these results. Should a more accurate and suitable parameter comparison be adopt to enhance recognition of the accuracy of the model? (E.g. Tg or other parameters)

  1. b) In section 3.2, the authors present the conclusion of "Adding C60 into cis-PI composites leads to a linear increase in density.". But in section 3.1 the authors also mention "One of the major contributors to this is the reduction in the degrees of freedom upon coarse-graining which causes large fluctuations in the simulation box volume compared to the UAMD simulations". So is the change of the density only due to the increase of the filler mass? Can it be explained by combining the changes in the box size before and after the model is balanced?

  1. c) The difference between the maximum and minimum values of R0 and Rg given in Table 2 are only 1.92% and 2.04% (calculate by: (Rmax-Rmin)/ Rmin), which are even smaller than the errors. Is the conclusion of "extension of cis-PI chains." reliable? Are there any other results to support this conclusion? (E.g. molecular chain configurations before and after balance)

  1. d) What kind of expression for error bar is adopted in all results in the manuscript? Are the maximum and minimum values? Or standard deviation? Or other forms of error expression?

  1. e) The curves in Fig.2(b) overlap severely. It is recommended to use partial magnification or other methods for more clear characterization.

  1. Please revise the manuscript carefully to improve readability.

Author Response

Dear reviewer,

The manuscript and supporting information were revised according to suggestions. Changes made in the revised manuscript were shown in red. We would like to acknowledge useful suggestions, which certainly helped to increase the quality of in the present form manuscript.

Please find out the response to your suggestion.

Round 2

Reviewer 1 Report

Publish as is.

Author Response

Reviewer Comment: Publish as is.

Response: Thank you for your supports.  The manuscript is carefully proofread.

Reviewer 3 Report

The authors have revised the introduction, the simulation content and the conclusion properly. The revised manuscript has the value for publication and potential guidance for relative research. Therefore, I recommend that this paper should be published as it is.

Author Response

Reviewer Comment:  I recommend that this paper should be published as it is.

Response: Thank you for your supports.  The manuscript is carefully  check the spell and gramma errors.